# Paediatric cancer burden in Namibia: A 10-year retrospective, analytical cohort study of patients admitted at Windhoek Central Hospital

**Ndapewa Ottilie Kaholongo** [1]*, **Runyararo Mashingaidze-Mano** [2]

1 School of Medicine, Faculty of Medicine and Veterinary Sciences, University of Namibia, Windhoek, Namibia, 2 Division of Paediatrics, Department of Maternal and Child Health, School of Medicine, Faculty of Medicine and Veterinary Sciences, University of Namibia, Windhoek, Namibia

* kndapewak@gmail.com

**Data Availability Statement:** All relevant data are within the paper and its supporting information files, and additional data can be found in the

## Abstract

### Background

Childhood cancers are known to cause significant morbidity and mortality, and the incidence has been increasing exponentially in developing countries. Two studies performed in Namibia in 1988 and 2010 have shown changes in the pattern of paediatric cancers over the years. There is a constant need to have updated statistics on the changing trends in the frequency of different types of cancers to inform policy hence the reason for the current study.

### Methods

An analytical retrospective cohort study was performed to analyse paediatric oncology cases that were admitted to the paediatric oncology unit (ward 8 west) at Windhoek Central Hospital (WCH) between 01 January 2011 and 31 December 2020. The study analysed the files of paediatric patients admitted with a paediatric cancer diagnosis from the age of 0 to 16 years. The research data was collected between July 2021 and September 2022.

### Results

A total of 174 paediatric cancer patient files met the inclusion criteria. Haematopoietic cancers were the most commonly occurring diagnosis of a paediatric cancer type in the study population (44.8%), of which leukaemias were the most common type of haematopoietic cancer. The other types of cancer apart from haematopoietic cancers consisted of embryonal cancers (37.9%), soft tissue and bone sarcomas (13.8%), and brain or CNS cancers (3.4%). The median age at diagnosis was 5.13 years, with an age range of 0 to 15 years. Fifty five point seven percent (55.7%) were males and 44.3% were females, with a male: female ratio of 1.26:1. Overall, most of the cancers were positively correlated with age, with the interactive-forward test indicating that the method of diagnosis and time significantly ($P < 0.05$) affected identification at the hospital.

following link: https://zenodo.org/badge/DOI/10.5281/zenodo.8270769.svg.

**Funding:** The authors received no specific funding for this work.

**Competing interests:** The authors have declared that no competing interests exist.

**Abbreviations:** CDC, Centers for Disease Control; CNS, Central nervous system; CT, Computed tomography; HIV, Human Immunodeficiency Virus; HIV/AIDS, Human Immunodeficiency Virus/Acquired Immune Deficiency Syndrome; MOHSS, Ministry of Health and Social Services; MRI, Magnetic Resonance Imaging; PMTCT, Prevention of Mother to Child Transmission; SPSS, Statistical Package for Social Sciences; Std, Standard; WCH, Windhoek Central Hospital; WHO, World Health Organization.

## Conclusions

Haematopoietic cancers remain most common type in Namibia. However, there has been a change in the ranking of the other childhood cancer subtypes over the last 3 decades. Good access to diagnosis and treatment modalities was noted as key to detection and clinical outcomes in the last 10 years (2011 to 2020). For future follow-up studies, prospective studies are recommended.

## Background

Childhood or paediatric cancers are known to cause significant morbidity and mortality and are the leading cause of death in children under 19 years of age [1]. Each year an estimated 400 000 children and adolescents of 0 to 19 years develop cancer [2]. Many of these cases (approximately 70%) occur in developing countries. While there are no age-specific cancers in the paediatric age group, there are certain cancers that present in specific childhood years [3]. The most common types of childhood cancers include leukemias, brain cancers, lymphomas and solid tumours, such as neuroblastoma and Wilms tumours [2, 4].

A study performed in Zimbabwe examined the pattern of paediatric cancers in children aged 0–14 years who were registered in the Zimbabwean Cancer Registry from 2000 to 2009 [5]. The study found that there was a prevalence of childhood cancer of 3.8% of all the malignancies recorded at the Zimbabwe National Cancer Registry over a ten-year period. The most common cancers were nephroblastoma, retinoblastoma and Kaposi sarcoma. Haematological cancers, specifically leukaemia and lymphomas, were also prevalent but were not the most common cancers [5].

Namibia a diverse nation of approximately 2.59 million people [6] had an estimated number 11 248 Namibians diagnosed with a malignant neoplasms in the last Namibian National Cancer Registry in 2014 [7]. Children under 15 years diagnosed with cancer constituted 3.3% [7]. However, it is known that in many African developing countries, such as Namibia, this estimate could be higher due to underdiagnoses or misdiagnosis of paediatric cancers [8]. In Africa, the incidence rates of some childhood cancers are much higher than those in developed countries [8].

Childhood cancer is curable [1], and in high-income countries, where comprehensive services are generally accessible, more than 80% of children with cancer are cured. In low- and middle-income countries (LMICs), less than 30% are cured [4, 9]. Stefan [8] suggests that the survival rate of paediatric cancers in Namibia was a mere 17%, in comparison to the higher rates in Western countries, such as the United States, where the survival rate is 85% [1]. Generic medicines can cure most of the childhood cancers as well as other forms of treatment, including surgery and radiotherapy. Treatment of childhood cancer can be cost-effective in all income settings [4]. In Africa however, despite the continent bearing a heavy burden of childhood cancer, more than 80% of children with cancer die without access to adequate treatment. Furthermore, only 29% of low-income countries report that cancer medicines are generally available to their populations compared to 96% of high-income countries [10]. In addition, avoidable deaths from childhood cancers in LMICs result from lack of diagnosis, misdiagnosis or delayed diagnosis, obstacles to accessing care, abandonment of treatment, death from toxicity, and relapse [4, 9].

Two studies performed in Namibia in 1988 and 2010 have shown changes in the pattern of paediatric cancers over the years [8, 11]. There is a constant need to have updated statistics on

the changing trends in the frequency of different types of cancers in order to drive continuous improvements in the quality of care, and to inform policy decisions [8, 12]. Little research has been done in the field of paediatric oncology in Namibia, similar to many other developing countries in Africa, and yet, worldwide, the majority of research in paediatric oncology comes from developed countries such as the United States [8]. This is counterintuitive to the fact that Africa carries a heavy burden of paediatric cancer. Therefore, there is a significant need for more research to be done in African developing countries such as Namibia. The last national cancer registry was reported in 2014, which included the total incidence of cancers in the whole Namibian population [7]. The aim of our study therefore was to document the paediatric cancer, identify factors associated with cancers diagnosis, diagnosis, treatment and long term outcome.

## Methods

A retrospective, analytical study of records of cancers in children aged 0–16 years registered and admitted at the Windhoek Central Hospital (WCH) from 01 January 2011 to 31 December 2020 was carried out. The hospital management at Windhoek Central hospital gave us permission to utilise their space for data collection. Namibia is a country located in southern Africa, which is divided into 14 regions, with Windhoek being the capital city, located in the Khomas region (Fig 1 shows the administrative regions on Namibia).

Windhoek Central hospital is the only national hospital which offers treatment of paediatric cancers to the public under the umbrella of the Ministry of Health and Social Services. The hospital is situated in the capital city of Namibia and caters for patients from all regions as far as more than 800 kilometers away. Additionally, the hospital has a total bed capacity of 855 patients including oncology patients (the total bed capacity of the oncology ward 8 West is 29). The paediatric oncology is manned by general paediatricians and medical officers and has no resident oncologist.

### Data recording and analysis

The research data was collected between July 2021 and September 2022. Information collected for each patient included patient demographics: age, sex, region or area of residency, date of diagnosis, type of cancer diagnosed, method of diagnosis, HIV status (exposed versus unexposed), outcome, date of discharge or death. Patients whose type of cancer was not recorded were excluded from the study. We employed a data-collecting template using a Microsoft Excel spreadsheet, which was used to collect the study variables, eliminating patients' identities.

The raw quantitative data on the Microsoft Excel spreadsheet for each variable were then analysed using the Statistical Package for Social Sciences (SPSS) version 29.0 statistical tool.

In this study, we characterized childhood cancers in Namibia with regard to the type of cancer diagnosed, sex and age distribution, age of diagnosis and date at discharge or death (whichever is applicable), region or area of residency, the impact of HIV (exposed versus unexposed), and eventual outcome of the patient. Eventual outcome consists of the treatment and/or management that the patient received before discharge or death (whichever is applicable). The method of diagnosis was also evaluated, and methods were grossly grouped into three main classifications: clinical (patient history and physical examination); imaging [ultrasound, radiology (X-ray, computed tomography (CT) scan, magnetic resonance imaging (MRI))]; and histology (biopsy). Files with missing or poorly recorded information with regards to the type of cancer diagnosed, diagnostic and treatment method were excluded from the study.

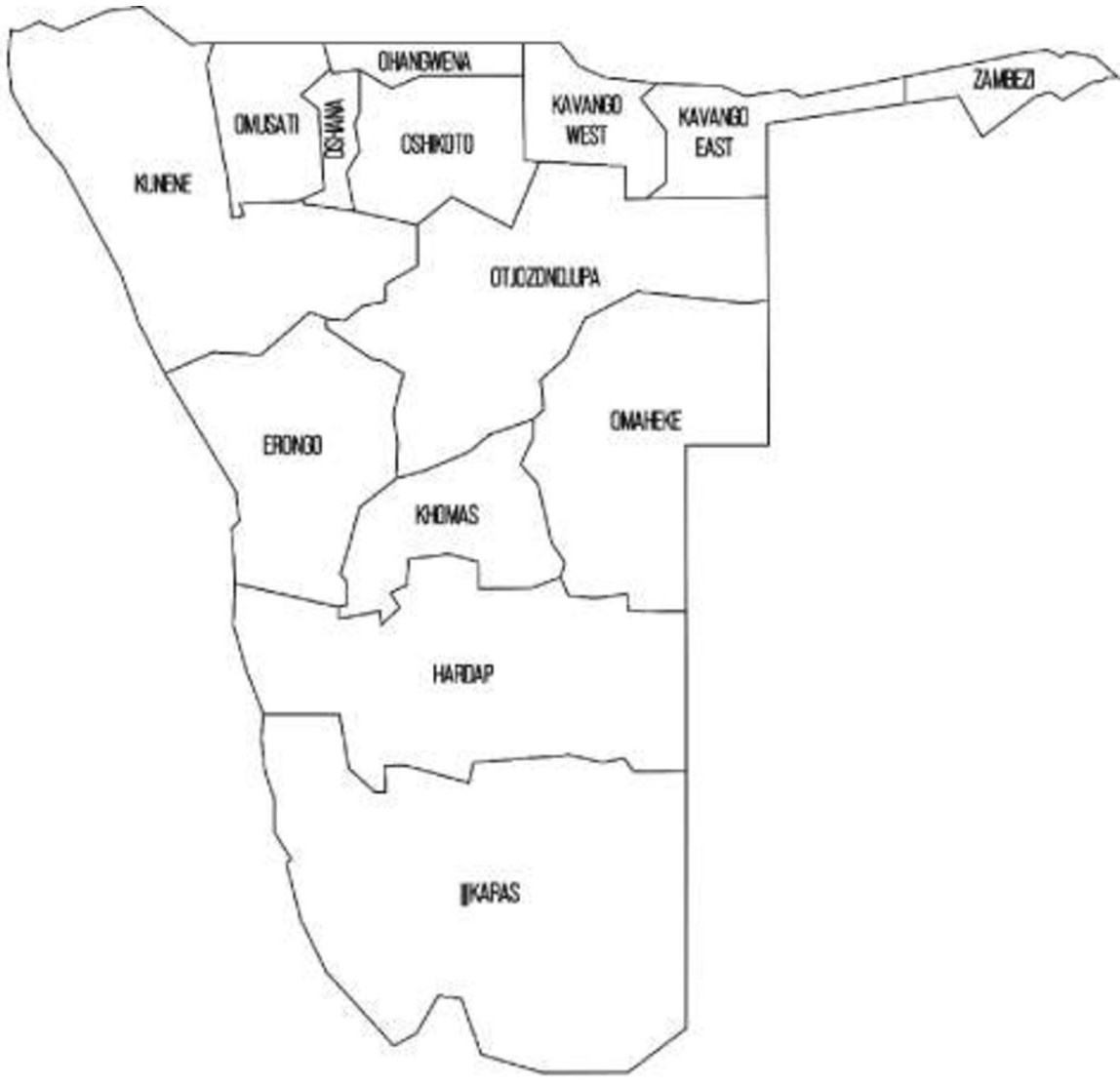

**Fig 1. Namibia's administrative regions [13].**

## Results

### Selection of the study sample

A total of 1000 paediatric oncology files were available for the study. Eight hundred and twenty-six (826) were excluded from the study for various reasons. Three hundred and sixty-three files had poorly recorded information, while 463 files were **not** admitted during the study period. Only 174 patient files met the study criteria (admitted from 01 January 2011 to 31 December 2020) and were included in the study (Fig 2).

### Demographic characteristics of the study participants

The median age at diagnosis was 5.0 years, with an age range of 1 to 15 years and a mean age of 5.13 years. Of the 174 patient files, 55.7% were males and 44.3% were females, with a male: female ratio of 1.26:1.

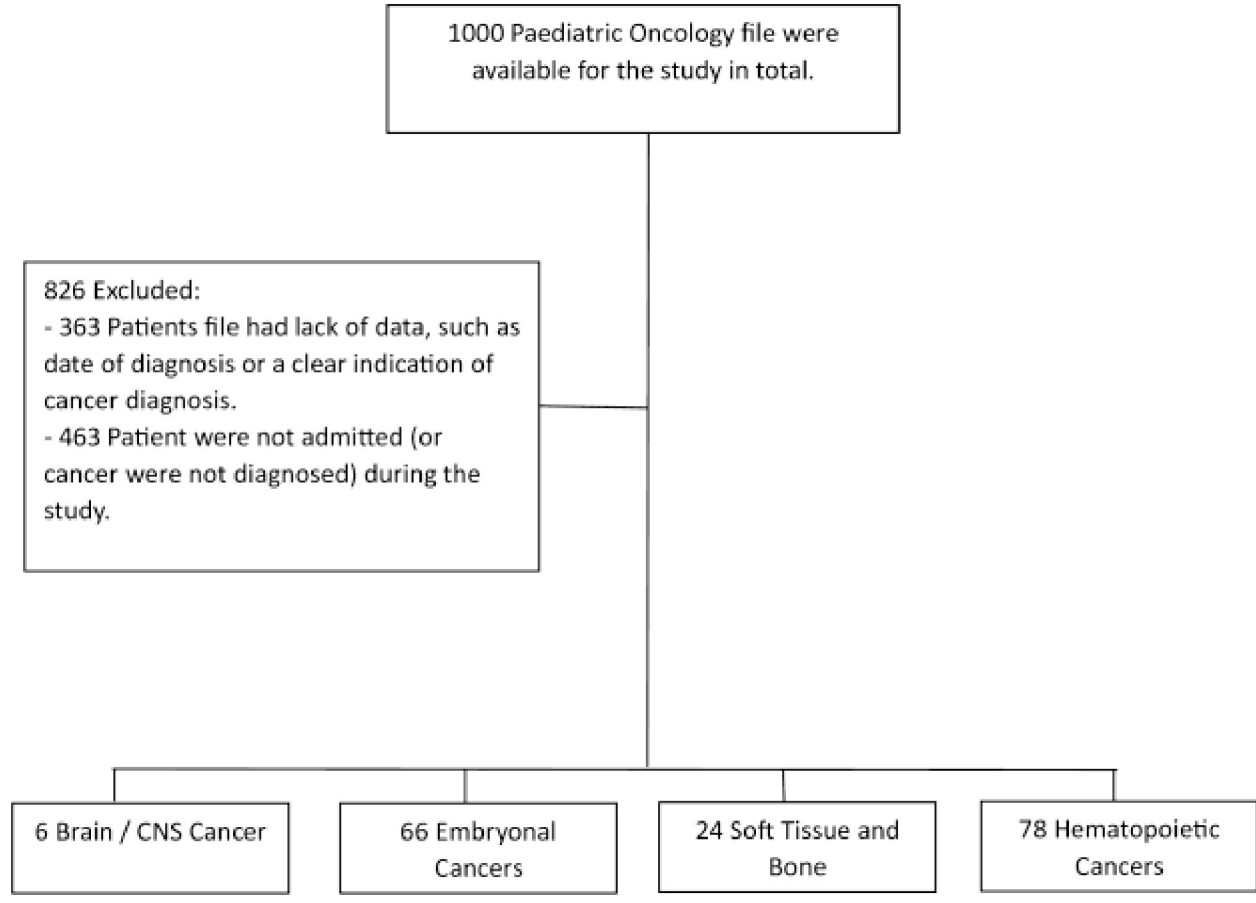

**Fig 2. A summarized flow chart of the patient files analysed for paediatric participants admitted at an oncology unit.**

### Region of residence

The Khomas region had the highest number of patient files admitted during the study period (20.7%), followed closely by the Oshana region (16.7%). The Kavango region contributed 10.9%, Erongo and Ohangwena contributed 9.8%, Oshikoto contributed 6.9%, Kunene contributed 6.3%, Otjozondjupa 5.2%, Omusati contributed 4.6%, and Zambezi contributed 4.0%. Karas, Hardap and Omaheke contributed the lowest number of admissions with 1.1%, 1.7% and 2.3%, respectively. Most pediatric cancer cases diagnosed originated from rural areas, 139 (79.3%), compared with 36 (20.7%) from urban areas. The Khomas region was considered the urban area, while the other 12 regions were grouped as the rural region or area, as those regions usually look after patients from rural areas, and Khomas has the only referring hospital for all oncology cases and caters for the urban area. 'Table 1' summaries the demographic characteristics of the study participants.

### Paediatric cancer types

Haematopoietic cancers were the most commonly occurring cancer type, constituting 78 (44.8%) of all 174 recorded patients in this study. Embryonal cancers were the second most common cancer type in children, with a frequency of 66 (37.9%). Soft tissue and bone sarcomas had an incidence of 13.8% of the 174 total patients in the study, while brain or central

**Table 1. Demographic characteristics of paediatric oncology patients admitted to Windhoek Central Hospital during the study period.**

| Main attribute | Sub-attribute | Percentage (%) |
|---|---|---|
| **Sex** | Male | 55.7 |
| | Female | 44.3 |
| **HIV Status** | Exposed | 6.3 |
| | Unexposed | 93.1 |
| | HIV Status unknown | 0.6 |
| **Major location of residence** | Urban* | 20.7 |
| | Rural* | 79.3 |
| **Specific region of residence** | Khomas | 20.7 |
| | Zambezi | 4.0 |
| | Otjozondjupa | 5.2 |
| | Kavango | 10.9 |
| | Erongo | 9.8 |
| | Oshana | 16.7 |
| | Ohangwena | 9.8 |
| | Oshikoto | 6.9 |
| | //Karas | 1.1 |
| | Hardap | 1.7 |
| | Omusati | 4.6 |
| | Omaheke | 2.3 |
| | Kunene | 6.3 |

*Urban refers to the Khomas Region, *Rural refers to all regions other than Khomas.

nervous system (CNS) cancers were the least commonly occurring with a mere 3.4% incidence. Fig 3 below illustrates the types of cancers diagnosed.

## Subtypes of paediatric cancer

Leukaemias were the most common cancer subtype (Table 2). Fifty-nine (33.9%) of the recorded patients were diagnosed with leukaemia, which is over one-third of the 174 patients recorded. Nephroblastomas, lymphomas, rhabdomyosarcomas, neuroblastomas and retino-blastomas were also very common, with frequencies of 29 (16.7%), 19 (10.92%), 17 (9.77%), 14 (8.05%), and 13 (7.47%), respectively, out of the 174 paediatric patients (Table 2). Astrocyto-mas, ependymal tumours, fibrosarcomas, Kaposi sarcomas, teratomas, germ cell tumours and yolk sac tumours were less common and together comprised 17 (9.76%). Glioma-pontine tumours, melanomas, osteosarcomas, squamous cell carcinomas, hepatoblastomas and dysger-minomas were rare in this study, each contributing 1 (less than 1% of all the cancers).

## Age distribution and paediatric cancer types

The distribution of pediatric oncology cancer was strongly related to age group (**Table 3**). There was an inverse relationship between cancer and age, as cancers were more common at younger ages and less common at older ages. There were significantly more cancers diagnosed in the age category of 0–5 years for each of the cancer types. Of the 174 admissions, 104 were in the age group of 0–5 years, constituting 59.8% of all cancers, 34.5% were in the age category of 6–10 years, and only 5.70% of all cancers were diagnosed in children aged 11–15 years. Of note, haematopoietic cancers were the most diagnosed cancer type in the age category 0–5 (the

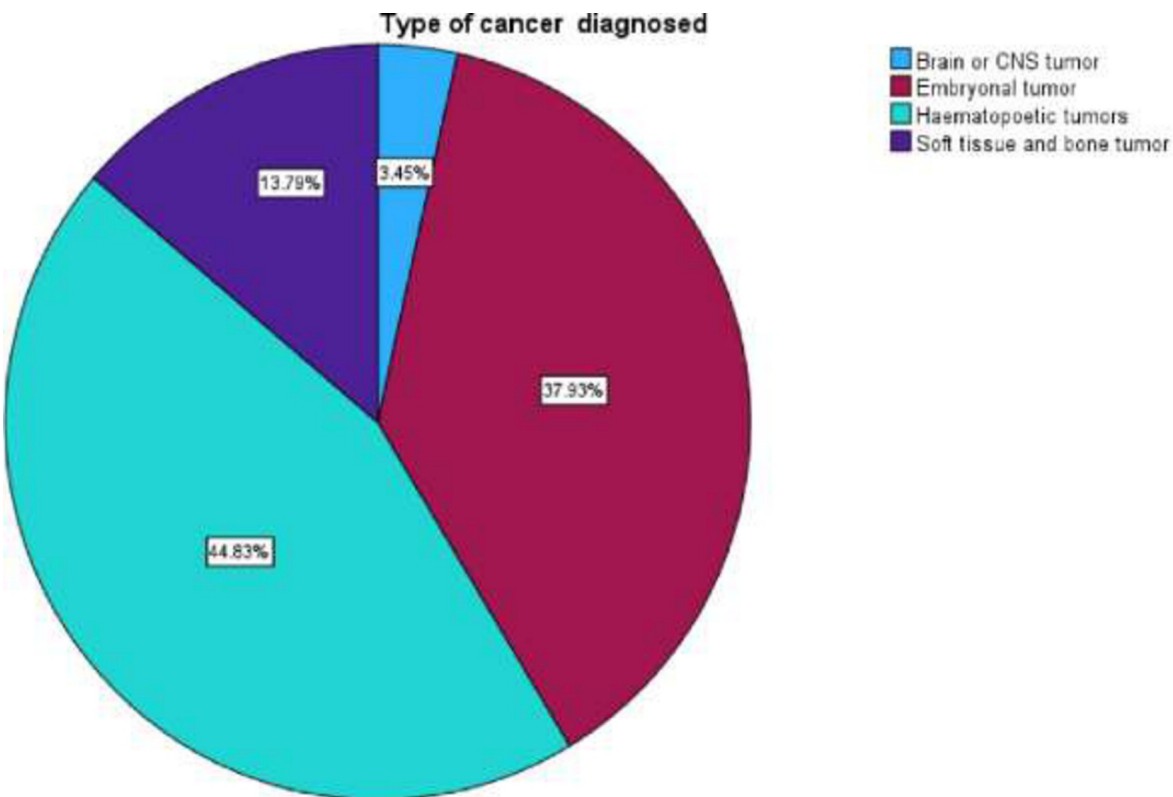

**Fig 3. Percentage of the types of paediatric cancer in paediatric patients admitted to an oncology unit.**

most commonly occurring age category overall), with a frequency of 44, constituting 25.3% of all the cancers recorded in this study. There were no soft tissue or bone sarcomas or brain or CNS cancers recorded in the age category of 11–15 years.

## Sex distribution and paediatric cancer type

Overall, males were more commonly diagnosed than females for most of the cancer types (Table 4). Males were twice as affected as females for brain or CNS cancers (4 versus 2) and almost twice as much for embryonal cancers (41 versus 25). For soft tissue and bone sarcomas, males were only slightly more affected than females, with a frequency of 13 within the cancer group itself, compared to 11 in females (Table 4). Haematopoietic cancers, the most common cancers in the whole study, occurred in equal proportions for both sexes, with a frequency of 39 for both sexes.

## Paediatric cancer type and HIV status

The HIV status of the participants was categorised either as HIV negative, HIV infected, HIV exposed uninfected (HEU) and unknown if the HIV status was not available. Among the participants, 81% were negative, 4% infected, 4.6% HEU and the remaining 10.3% had an unknown status. HIV exposure whether HEU or infected was experienced in all the cancer types. However, all the participants with brain or CNS tumour had the HIV status available in contrast to the other 3 cancer groups (Table 5).

**Table 2. Percentage of the subtypes of paediatric cancers in patients admitted to a paediatric oncology unit.**

| Paediatric cancer type | Paediatric cancer subtype | Valid Percent (%) |
|---|---|---|
| Haematopoietic cancers | Leukaemias | 33.9 |
| | Lymphomas | 10.9 |
| Brain or CNS cancers | Astrocytomas | 2.3 |
| | Ependymal tumours | 0.6 |
| | Glioma-pontine tumours | 0.6 |
| Soft tissue and bone sarcomas | Rhabdomyosarcomas | 9.8 |
| | Fibrosarcomas | 0.6 |
| | Kaposi sarcomas | 1.1 |
| | Melanomas | 0.6 |
| | Osteosarcomas | 0.6 |
| | Squamous cell carcinomas | 0.6 |
| Embryonal cancers | Nephroblastomas | 17.2 |
| | Neuroblastomas | 7.5 |
| | Retinoblastomas | 7.5 |
| | Teratomas | 2.3 |
| | Germ cell tumours | 1.2 |
| | Yolk sac tumours | 1.1 |
| | Hepatoblastomas | 0.6 |
| | Dysgerminomas | 0.6 |
| | **Total** | ***100.0** |

*Rounded off to the nearest whole number

## Method of diagnosing cancer

Histology using biopsies was the most widely used method of diagnosis, constituting 72.4% of all diagnosed cancers. Occasionally, imaging [ultrasound, radiology (X-ray, CT scan, MRI)] was also used in conjunction with histology (21.3%). Four percent of the participants had diagnoses done through use of imaging alone. In a few cases, the diagnosis was made based on a combination of clinical assessment, imaging, and histology (biopsy) (2.3%). Histology included tissue biopsies, cytology specimens and the bone marrow aspirates and biopsies. Almost all the haematopoetic tumours were diagnosed solely with histology. For the embryonal and soft tissue tumours histology was the main diagnostic modality followed by a combination of imaging and histology. In contrast all the brain or CNS tumours were diagnosed either by imaging or a combination of imaging and histology (Fig 4). A detailed figure with more information on the diagnostic method and type of cancer diagnosed is attached as S1 File.

**Table 3. Age distribution of paediatric cancers according to 3 age categories for paediatric cancer participants admitted at an oncology unit.**

| % Type of cancer diagnosed | Age Group in years | | | Total |
|---|---|---|---|---|
| | **0–5** | **6–10** | **11–15** | |
| Brain or CNS cancer | 2.9 | 0.6 | 0 | 3.5 |
| Embryonal cancer | 22.4 | 11.5 | 4 | 37.9 |
| Soft tissue and bone sarcoma | 9.2 | 4.6 | 0 | 13.8 |
| Haematopoietic cancer | 25.3 | 17.8 | 1.7 | 44.8 |
| **Total** | 59.8 | 34.5 | 5.7 | **100.0** |

**Table 4. Sex distribution of the paediatric cancers in patients admitted to a paediatric oncology unit.**

| Type of Cancer diagnosed | Sex | | |
|---|---|---|---|
| | **Male (%)** | **Female (%)** | |
| Brain or CNS cancer | 2.1 | 2.3 | |
| Embryonal cancer | 33 | 19.5 | |
| Soft tissue and bone sarcoma | 15.5 | 5.2 | |
| Haematopoietic cancer | 49.5 | 17.2 | |
| Total% | 55.8 | 44.2 | **100** |

## Factors associated with cancer types

Overall, most of the cancers were positively correlated with age, while some were correlated with location and HIV status. A few cancers were also combinely correlated with gender, time and method of diagnosis. Age was the most influential variable, followed by gender and location, while HIV had little impact on cancers (Fig 5).

Fig 5 below is a detrended analysis diagram, of the cancer type and environmental variables.

The interactive-forward test indicated that the method of diagnosis and time significantly ($P < 0.05$) affected identification at the hospital (Table 6).

## Treatment modalities used for the cancer patients

The majority of paediatric patients (44.3%) were treated solely using chemotherapy. This was followed by 16.1% of patients who were treated with both chemotherapy and surgery. Of note, these are patients who received either neoadjuvant chemotherapy or postoperative chemotherapy or both, depending on the specific type of cancer and individual cases. The patients who ended up in palliative care in addition to their primary form of treatment and/or management together constituted 21.3%, which is approximately a quarter of the total recorded patients. The patients who received radiotherapy constituted a tenth of all the patients (14.4%). Of significance, 31% of the patients underwent surgery to remove a tumour, which is one-third of the total recorded patients.

Additionally (Fig 6), the treatment modalities for the different cancers was based on the type of cancer diagnosed. For the haematopoetic cancers 84.6% received chemotherapy, 14.3% both chemotherapy and radiotherapy with the remaining requiring palliative care and or surgery for those with lymphomas. Patients diagnosed with brain or CNS cancers were treated with either radiation alone (33.2%), surgery alone (16.7%), supportive care alone (16.7%) or a combination of surgery and supportive care (16.7%), chemotherapy and radiotherapy (16.7%).

For those patients with embryonal tumours, all the treatment modalities were used except for a combination of either surgery with supportive care or surgery with radiotherapy. In the same vein, patients with soft tissue and bone tumours required all the other treatment modalities except solely radiotherapy or surgery and a combination of either chemotherapy surgery plus supportive care and chemotherapy, surgery and radiotherapy. For more information a detailed S2 File is attached.

## Length of hospital stay

Participants were divided into three groups (Table 7); according to length of hospital stay (LOS): short (500 days and below), medium (501–1000 days) and long (1001 days and above). Approximately 1.7% of participants were long-term inpatients, compared to 4.6% medium and 93.7% short-term stay. LOS had a mean of 218.6 days and a median of 185.5 days, IQR

**Table 5. Distribution of paediatric cancers according to HIV status for participants admitted at an oncology unit.**

| Type of Cancer diagnosed | % HIV status | | | | Total |
|---|---|---|---|---|---|
| | HIV Exposed uninfected | HIV infected | HIV negative | ^Unknown | |
| Brain or CNS tumor | 0.6 | 1.1 | 1.7 | 0 | 3.4 |
| Embryonal | 1.1 | 0.6 | 32.8 | 3.4 | 37.9 |
| Haematopoetic | 2.3 | 1.1 | 36.2 | 5.2 | 44.8 |
| Soft tissue and bone tumor | 0.6 | 1.1 | 10.3 | 1.7 | 13.8 |
| Total | 4.6 | *4.0 | 81 | 10.3 | *100 |

*Rounded off to the nearest whole number; ^The HIV status was not recorded in the files.

(102.5–252 days). The LOS ranged from 1to 1599 days (over 51 weeks or 4.3 years. Fig 7 is an illustration of the link between cancer type diagnosed and length of hospital stay. A S3 File is attached for further information.

## Treatment outcome

The treatment outcome was either death or discharge. The mortality rate for all the participants was 17.2% (Fig 8). Among the participants who died 46.7% had embryonal tumours, 30% soft tissue and bone tumours with the remaining 23.3% haematopoietic tumours.

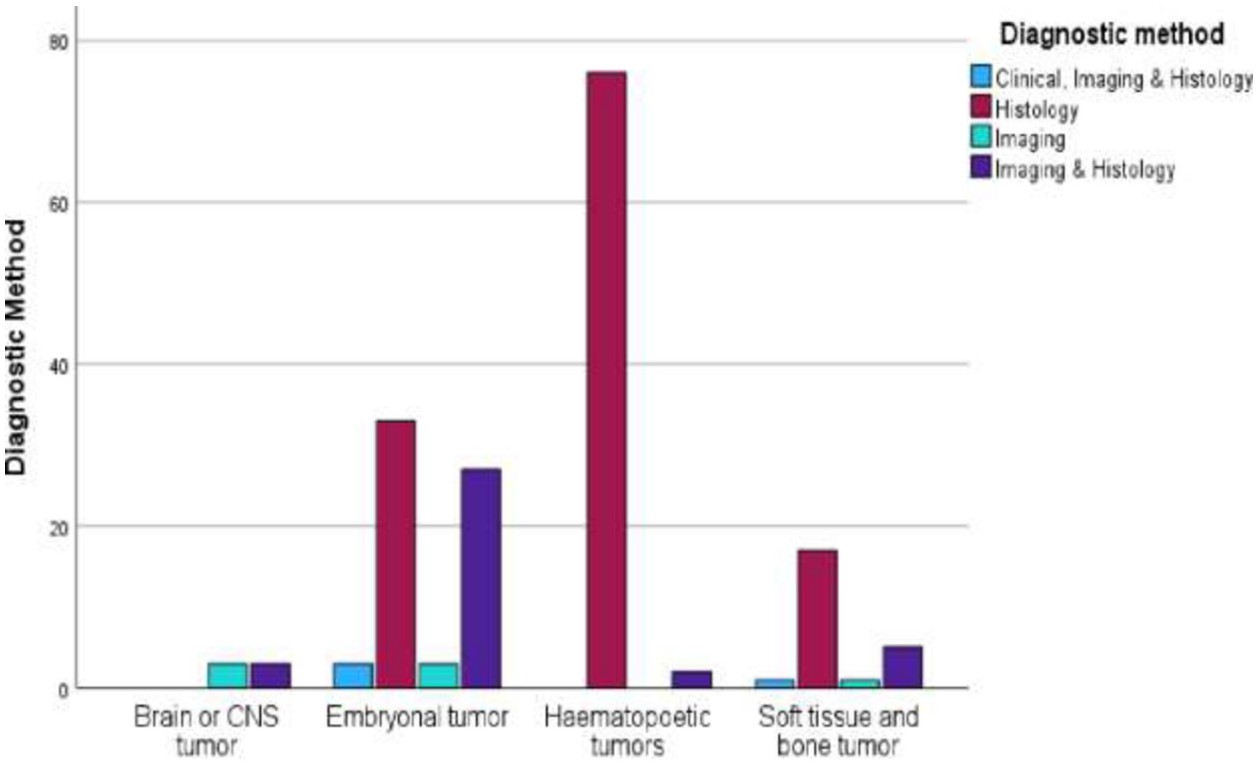

**Fig 4. Type of cancer diagnosed and the diagnostic method for participants admitted at an oncology unit.** *Histology (biopsy) includes tumour biopsies, cytology specimens, and bone marrow biopsies.

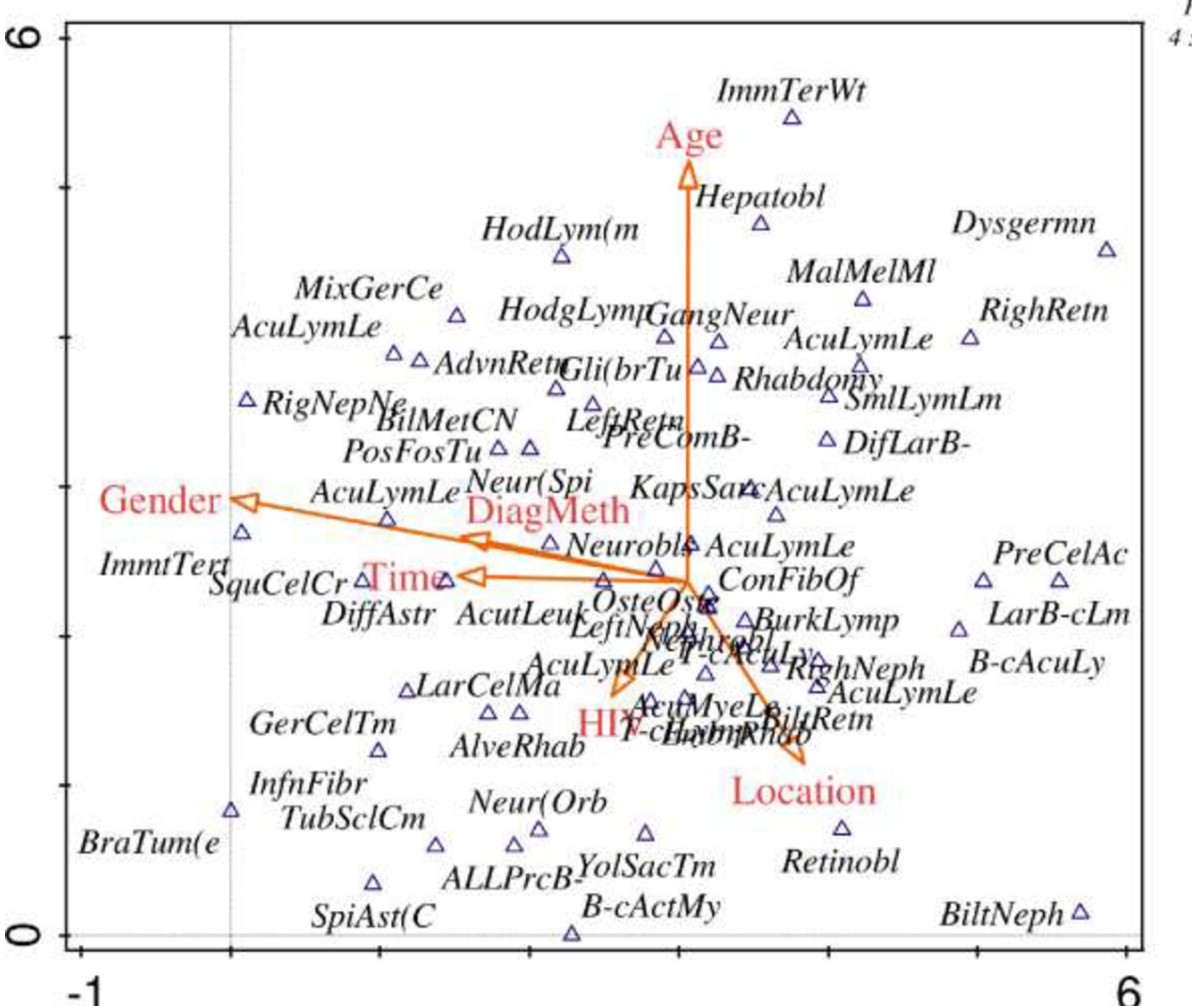

**Fig 5. Detrended correspondence analysis (DCA) diagrams of cancer type and environmental variables.** The direction of the vectors indicates the direction of maximum change of that variable, whereas the length of the vector represents how well the parameter explains the distribution of the data. External variable (Age = Age of patient, Gender = gender of the patient, Time = year in which the child was diagnosed, DiagMeth = Method used for diagnosis, HIV = HIV status of the patient, Location = Area the patient lived).

**Table 6. Detailed results of forward selection test analysis for paediatric participants admitted at an oncology unit.**

| External variables | Explains % | Contribution % | pseudo-F | P |
|---|---|---|---|---|
| DiagMeth | 1.1 | 25.3 | 1.9 | 0.002 |
| Time | 0.8 | 19.1 | 1.4 | 0.002 |
| Gender | 0.7 | 16.1 | 1.2 | 0.052 |
| Age | 0.7 | 15.9 | 1.2 | 0.106 |
| HIV | 0.6 | 12.8 | 1.0 | 0.582 |
| Location | 0.5 | 10.7 | 0.8 | 0.94 |

**Age** = age of patient, Gender = gender of the patient, Time = year in which the patient was diagnosed, **DiagMeth** = Method used for diagnosis, HIV = HIV status of the patient, Location = area the patient lived

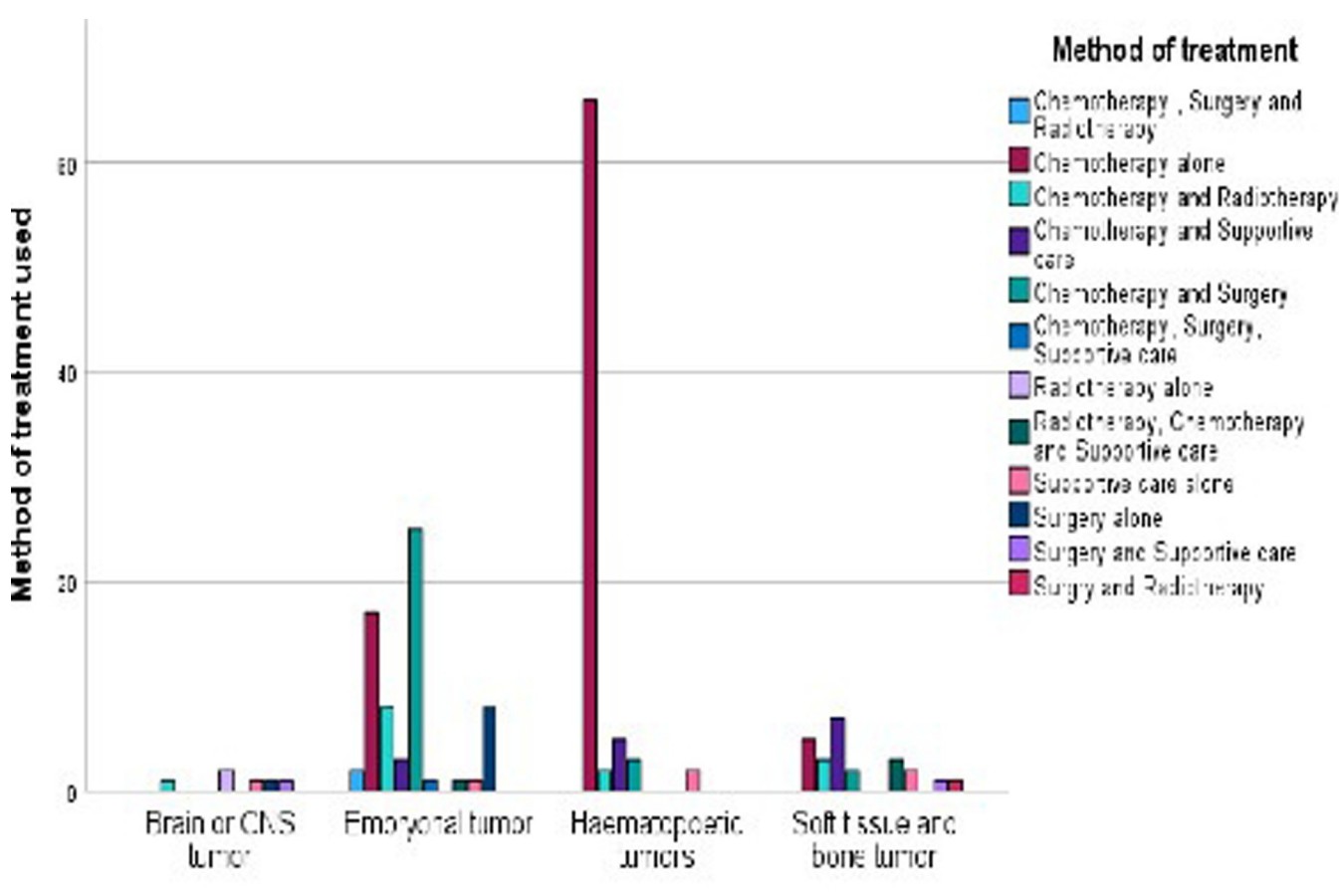

**Fig 6. Type of cancer diagnosed and the treatment modalities for participants admitted at a paediatric oncology unit.**

Furthermore, the participants died either before, during or after treatment and/or management, but all the patients died after the diagnosis was made.

## Discussion

This study examined the paediatric cancer pattern over a 10-year period. Males constituted more than half of the study participants (55.7%), while females constituted 44.3%. The ratio of males to females was 1.26:1. In general, it is known that male children are at greater risk of

**Table 7. Type of cancer diagnosed and length of hospital stay for participants admitted at a paediatric oncology unit.**

| Type of Cancer diagnosed | *Length of Hospital Stay | | | Total |
|---|---|---|---|---|
| | Long | Medium | Short | |
| Brain or CNS tumours | 0 | 0 | 6 | 6 |
| Embyronal tumours | 2 | 5 | 59 | 66 |
| Haematopoetic tumours | 1 | 2 | 75 | 78 |
| Soft tissue and bone tumour | 0 | 1 | 23 | 24 |
| **Total** | 3 | 8 | 163 | 174 |

Short = less than 500 days, Medium 501-1000days, Long = ≥ 1001 days

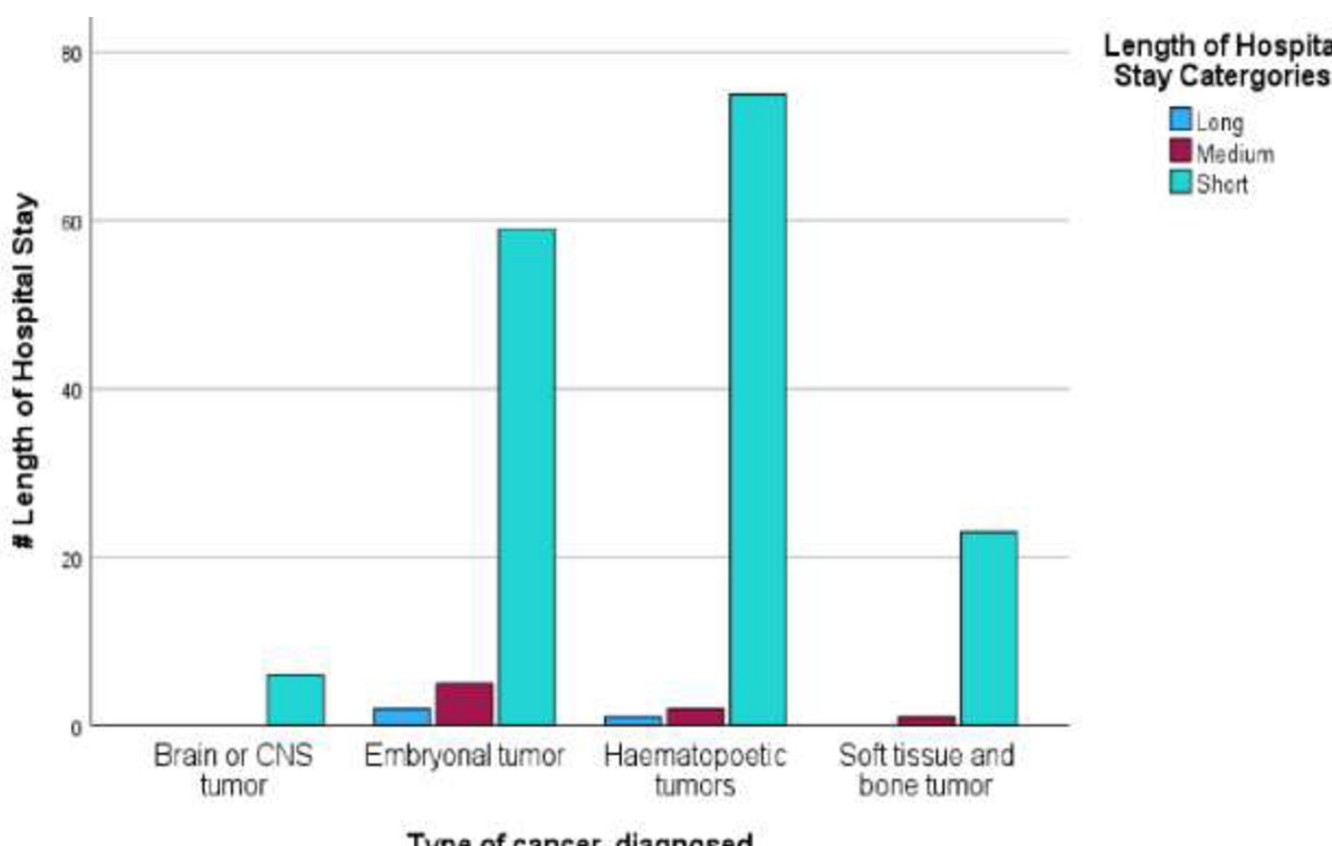

**Fig 7. Type of cancers diagnosed and length of hospital stay for participants admitted at a paediatric oncology unit.** *Short = ≤ 500 days, Medium 501–1000 days, Long ≥1001 days.

developing cancer than their female counterparts [14]. However, the exact mechanisms that underlie this difference in childhood cancer incidence by sex are mostly unknown [15]. This may be because male children are often of higher birthweight, contract a higher number of childhood infections, and their pubertal growth is much more accelerated with a completely different hormonal profile compared to females [15]. These risk factors may contribute to the higher incidence of cancer in male children. These findings were similar to previous studies in

| Variable | Frequency | Valid Percentage |
|----------|-----------|------------------|
| Died | 20 | 17,2 |
| Alive | 144 | 82,8 |
| **Total** | 174 | 100 |

**Fig 8. Treatment outcome for participants admitted at a paediatric oncology unit.**

which male children were predominantly reported to have cancer compared to female children [5, 8, 11].

In this study, cancers were strongly related to age, as more cancers were found in the younger age groups. These findings are comparable to previous studies in both developing and developed countries that reported the highest incidence of cancer in children under 5 years of age [5, 8, 11, 16]. There is very little research to explain why children under 5 years of age develop cancer, but certain cancers tend to occur preferentially in infants, children and adolescents [3]. The majority of childhood cancers have no known cause with only about 10% having an underlying genetic cause. However, certain chronic infections, such as HIV, malaria and Epstein–Barr virus, are risk factors for childhood cancers [17, 18]. Additionally infants are more susceptible to infections than older children and adolescents due to their immature immune systems [19].

About 8.6% of our participants were exposed to HIV with a positive rate of 4.0%. Routine testing for HIV infection in all patients in Namibia presenting with cancer was introduced since 2009 [8]. The HIV infection was not limited to any group of type of cancer diagnosed. In contrast the study performed by Chitsike et al. [5] showed a higher incidence of HIV-associated Kaposi sarcoma. In that study, Kaposi sarcoma was the second most common cancer in children, accounting for 15.7% of all cases. However, HIV status was not mentioned in an earlier study performed in 1988 by Wessels [11]. This is presumably because HIV is still a relatively new disease, as it was first discovered in 1983 [20].

Most of the cases in this study originated from the rural areas of Namibia (79.3%). This is most likely due to many parents seeking alternative medicine when their child is ill [8]. Further studies are encouraged to investigate the risks and benefits of alternative medicine for diagnosing paediatric cancers, as alternative medicine is an important aspect of our society. There is also the failure of healthcare workers to diagnose a malignancy because of inexperience in recognizing a childhood malignancy or because of the difficulty in distinguishing between the overlapping symptomatology of cancer and commonly occurring infectious conditions such as malaria and tuberculosis [11]. However, other studies show that urban–rural differences in cancer incidence and trends vary across the world. A study performed in Sudan [21] found that approximately 76% of Sudanese children diagnosed with cancer lived in rural areas. Another reason for the high frequency of patients coming from rural regions found in our study could be that patients in rural areas are often exposed to cancer-associated modifiable risks such as HIV, chronic childhood infections, and secondary smoking although this may need further probing [12].

In our study the commonly occurring cancers were leukaemia, nephroblastomas, lymphomas, retinoblastomas, and rhabdomyosarcomas. However, there were some notably differences from previous studies performed in Namibia in which case, leukaemias increase one and half times in our study compared to previous study. In addition the second most occurring cancer, nephroblastomas in our study was different from a study by Stefan in which case retinoblastomas was the second [8]. Of note, there were some specific cancers not mentioned by Stefan such as, rhabdomyosarcomas and neuroblastomas.

When looking closely at the cancer subtypes, there was a considerable change seen in the ranking compared to previous studies [8, 11]. In this study, there were many childhood cancers that decreased since the study by Stefan [8] 12 years ago, but leukaemias doubled. This is most likely due to better methods of diagnoses of childhood cancers.

In another study by Chitsike et al. [5], leukaemias in general were not as common (16.2%) in comparison to our study (33.9%). Nephroblastoma or Wilms tumour was the most commonly occurring cancer in children in Zimbabwe, in contrast to leukaemias for Namibia. This was followed by Kaposi sarcoma (15.7%), retinoblastoma (13.1%), non-Hodgkin's lymphoma

(10.3%), leukaemia (8.9%), brain or nervous tissue (6.1%), connective tissue (5.9%), bone (5.5%), Hodgkin's lymphoma (3.2%), non-melanoma skin cancer (1.9%), and miscellaneous cancers (13.2%) (5). This is most likely due to a difference in the incidence of HIV between Namibia and Zimbabwe, resulting in the difference in trends and availability of diagnostics which have not been reported in the later.

It has been reported that paediatric cancer can be curable if access to diagnosis and early treatment is offered in all income settings. In the low and middle income countries however avoidable deaths from childhood cancer occur due to lack of diagnosis, misdiagnosis or lack of diagnosis [4]. In our study although we could not report on the time of the diagnosis versus the cancer stage all our participants had access to almost all the diagnostic methods. Histology which included bone marrow biopsies were performed for all the Leukaemias. The availability of the diagnostics like CT or MRI scan was commendable in our study.

Most childhood cancers can be cured with generic medicines and other forms of treatment, including surgery and radiotherapy [4]. In this study patients had access to several modalities of treatment including chemotherapy, radiotherapy, surgery and combination where necessary. However, some patients ended up requiring supportive or palliative care. The patients who ended up in palliative care in addition to their primary form of treatment and/or management together constituted 21.3%, which is approximately a quarter of the total recorded patients. This is significant, as it may imply that patients present late when the cancer has already progressed. Other studies in Namibia and Zimbabwe did not report on the cancer treatment modalities [5, 11]. Stefan [8] however reported that all brain and spinal tumours were treated with radiology but does not mention other treatment modalities.

In our study, we had a mean or average length of stay at Windhoek Central Hospital of 281.6 days or at least 9 months. The longest stay for 1 patient at the Paediatric Oncology unit was 1599 days or at least 4 years. Merill [22] found that compared to the average stay among children and adolescents without cancer, those for cancer care were more than twice as expensive and slightly longer, which has implications on the financial costs incurred by the hospital. In our study the patients stayed for long in hospital more than average mostly likely due to the places of origin which can be as far as more than 800 kilometers away. As a result patients stay at the treating hospital until completion of treatment. Previous studies performed in Namibia and Zimbabwe [5, 8, 11] did not report on this variable of the length of hospital stay or the treatment outcome.

Cure rate of cancers in LMICs is around 30% in contrast to the high income countries where cure rate can be as high as 85% [4]. In our study, we had a mortality rate of 17.2%. This is comparable with the United States, where the mortality rate is approximately 15% [23]. In contrast, in the European region, the mortality rate ranges from 9% to 57% [1]. This is in contrast to Africa in general, where the survival rate of children with cancer [11] is approximately 20% [24]. This could be due to reduced availability of resources and late presentation. Despite our participants being managed by general paediatricians the survival rate was high. This in contrast to a study in Zambia where the presence of a resident oncologist did not affect the completion of treatment for oncology patients [25].

Our study was a retrospective study, and some files had missing information and were therefore excluded from the study. Therefore, the recommendation is for future studies in paediatric oncology in Namibia to be prospective instead of retrospective, which will ensure that the required information is captured adequately. In addition, the study did not report on the clinical stages of the various cancers that could then be compared with treatment outcome and modalities used. The data captured is only for the public institution hence excluding private sector which may produce different results.

## Conclusion

In conclusion, hematopoietic cancers were the number one most common cancer in children over the ten year period, from 01 January 2011 to 31 December 2020 in Namibia. The ranking of the other paediatric cancer types and subtypes has changed since the last study that was done in 2010, with the exception of leukaemias, which have remained the most common cancer type in children since 2010. Overall, most of the cancers were positively correlated with age, while some were correlated with location and HIV status The interactive-forward test indicated that the method of diagnosis and time significantly (P < 0.05) affected identification at the hospital. The was good access to diagnostic and treatment modalities for the participants and the overall treatment outcome comparable to high income countries. We recommend that prospective studies be performed rather than retrospective studies in the future, as the majority of patient files consisted of poor documentation of information. A cancer registry is needed in the future to determine the true incidence of childhood cancers in Namibia. Additionally, there is need to upgrade some of the hospitals in other regions equipping them with proper diagnostic and treatment modalities to reduce length hospital stay and the distances patient travel to seek treatment at the only Central hospital in Namibia and Southern Africa at large.

## Supporting information

**S1 Checklist. STROBE checklist.**
(DOCX)

**S2 Checklist. PLOSONE clinical studies checklist.**
(DOCX)

**S1 File. Type of cancer in comparison with diagnostic method.**
(DOCX)

**S2 File. Method of treatment * type of cancer diagnosed crosstabulation.**
(DOCX)

**S3 File. % type of cancer diagnosed compared with length of hospital stay.** Length of hospital stay and cancer type frequencies and percentages for paediatric participants admitted at an oncology ward.
(DOCX)

**S4 File. Initial letter of approval to commence with the research.**
(PDF)

**S5 File. Letter of approval to publish the research from the Ministry of Health and Social Services (MOHSS).**
(PDF)

## Acknowledgments

Many thanks go to the assistant lecturer at the Department of Radiology, University of Namibia School of Allied Health Sciences, for his expert and valued statistical input. We would like to express our great felt thank you to Dr Tonny P. Tauro who helped in part of the analysis and copyediting the manuscript.

## Author Contributions

**Conceptualization:** Runyararo Mashingaidze-Mano.

**Data curation:** Ndapewa Ottilie Kaholongo.

**Investigation:** Ndapewa Ottilie Kaholongo.

**Project administration:** Ndapewa Ottilie Kaholongo.

**Software:** Ndapewa Ottilie Kaholongo.

**Supervision:** Runyararo Mashingaidze-Mano.

**Visualization:** Runyararo Mashingaidze-Mano.

**Writing – original draft:** Ndapewa Ottilie Kaholongo.

**Writing – review & editing:** Ndapewa Ottilie Kaholongo, Runyararo Mashingaidze-Mano.

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
