## [Decision Letter · Decision Letter 0]

21 Jun 2023

PONE-D-23-09242Paediatric cancer burden in Namibia, A 10 year retrospective, analytical study of patients admitted at Windhoek Central Hospital.PLOS ONE

Dear Dr. Kaholongo,

Thank you for submitting your manuscript to PLOS ONE. After careful consideration, we feel that it has merit but does not fully meet PLOS ONE’s publication criteria as it currently stands. Therefore, we invite you to submit a revised version of the manuscript that addresses the points raised during the review process.

We look forward to receiving your revised manuscript.

Kind regards,

Nontuthuzelo Iris Muriel Somdyala, Ph.D

Academic Editor

PLOS ONE

Journal Requirements:

5. We note that Figure 1 in your submission contain map images which may be copyrighted. All PLOS content is published under the Creative Commons Attribution License (CC BY 4.0), which means that the manuscript, images, and Supporting Information files will be freely available online, and any third party is permitted to access, download, copy, distribute, and use these materials in any way, even commercially, with proper attribution. For these reasons, we cannot publish previously copyrighted maps or satellite images created using proprietary data, such as Google software (Google Maps, Street View, and Earth). For more information, see our copyright guidelines: http://journals.plos.org/plosone/s/licenses-and-copyright.

Reviewers' comments:

Reviewer's Responses to Questions

**Comments to the Author**

1. Is the manuscript technically sound, and do the data support the conclusions?

Reviewer #1: Partly

2. Has the statistical analysis been performed appropriately and rigorously? 

Reviewer #1: No

3. Have the authors made all data underlying the findings in their manuscript fully available?

Reviewer #1: Yes

4. Is the manuscript presented in an intelligible fashion and written in standard English?

Reviewer #1: No

5. Review Comments to the Author

Reviewer #1: REVIEW REPORT

Title: Title: Paediatric cancer burden in Namibia, A 10 year retrospective, analytical cross-sectional study of patients admitted at Windhoek Central Hospital.

General comments:

Specific comments:

Abstract:

The abstract has all the sections required for an abstract. The background should be revised, this study is focused on paediatric cancer burden and not on the scarcity of research in paediatric oncology. The background focus should be on paediatric cancer burden

Method is not cross-sectional analysis.

Background:

Background need to be revised, the focus is on paediatric cancer, but there is repetition of ideas, the last paragraph should be included in the first part of this abstract. The second paragraph is providing justification for carrying out this study, therefore it should be the last and paragraph in the background which should also indicate the objective of carrying out this study.

There are redundant statements which need to be omitted or revised to show their relevance. A good example of this is this statement ‘Namibia is a diverse nation of approximately 2.59 million people’

The order of references in this section is awkward, the references need to be arranged in ascending sequence.

Method:

This section need to be revised and more information are required, the following information need to be clarified

When was data collected, exact months and year for this activity?

There are should be clarification of the ethical approval for this study.

o Which Review board provided ethical approval for this study?

o This was a retrospective study, why did the MOHSS and matron give consent for this study? Why is it stated that there was a waiver of consent which was given.

The setting for this study should be briefly described, who is providing care for this children with oncology, how big is the unit, how many children are admitted at one time. These information are useful to give context to the reader of this work.

Results

The data presented shows a clear spectrum of paediatric cancers in Namibia. The following should be added or reorganized for clarity and improvement of results

Data on histology should be included in the results especially for solid organ cancers.

Method of diagnosing cancers should be based on categories of cancers. It is very difficult to understand and make sense out of the results. It will be interesting to know how the diagnosis was made for brain tumours, haematologic cancers etc.

Treatment should be categorized for each group of tumours

Hospital stay may also be provided for each group of cancers as well eg. Haematologic cancers.

When reporting data use only a single decimal point.

Is there any relevance of showing regions from which patients came from in this report?

Discussion

In the discussion the authors are using their data to conclude on differences between ages and sex with occurrence of cancers. This is not possible as in the analysis and results no statical analysis was carried out. This is a descriptive study for all variables related to cancer, and no analysis was carried out and therefore no difference was observed.

6. PLOS authors have the option to publish the peer review history of their article (what does this mean?). If published, this will include your full peer review and any attached files.

Reviewer #1: No

---

## [Author Response · Author response to Decision Letter 0]

4 Sep 2023

The authors have responded to the specific reviewers' comments in the document labelled "Response to Reviewers".

---

## [Editor Report · Decision Letter 1]

28 Sep 2023

Paediatric cancer burden in Namibia, A 10 year retrospective, analytical cohort study of patients admitted at Windhoek Central Hospital.

PONE-D-23-09242R1

Dear Dr. Kaholongo,

We’re pleased to inform you that your manuscript has been judged scientifically suitable for publication and will be formally accepted for publication once it meets all outstanding technical requirements.

Kind regards,

Nontuthuzelo Iris Muriel Somdyala, Ph.D

Academic Editor

PLOS ONE

Additional Editor Comments (optional):

There are more edits made to enhance the quality of your paper. Please attend to edits and after that your manuscript will be ready and accepted.
---

## [Editor Report · Acceptance letter]

26 Oct 2023

PONE-D-23-09242R1 

Paediatric cancer burden in Namibia: A 10-year retrospective, analytical cohort study of patients admitted at Windhoek Central Hospital. 

Dear Dr. Kaholongo:

I'm pleased to inform you that your manuscript has been deemed suitable for publication in PLOS ONE. Congratulations! Your manuscript is now with our production department. 

Kind regards, 

on behalf of

Dr. Nontuthuzelo Iris Muriel Somdyala 

Academic Editor

PLOS ONE